# Targeting Breast Cancer: An Overlook on Current Strategies

**DOI:** 10.3390/ijms24043643

**Published:** 2023-02-11

**Authors:** Domenico Iacopetta, Jessica Ceramella, Noemi Baldino, Maria Stefania Sinicropi, Alessia Catalano

**Affiliations:** 1Department of Pharmacy, Health and Nutritional Sciences, University of Calabria, 87036 Arcavacata di Rende, Italy; 2Laboratory of Rheology and Food Engineering, Department of Information, Modeling, Electronics and System Engineering (D.I.M.E.S.), University of Calabria, 87036 Arcavacata di Rende, Italy; 3Department of Pharmacy-Drug Sciences, University of Bari “Aldo Moro”, 70126 Bari, Italy

**Keywords:** breast cancer, chemotherapy, endocrine therapy, therapeutic targets

## Abstract

Breast cancer (BC) is one of the most widely diagnosed cancers and a leading cause of cancer death among women worldwide. Globally, BC is the second most frequent cancer and first most frequent gynecological one, affecting women with a relatively low case-mortality rate. Surgery, radiotherapy, and chemotherapy are the main treatments for BC, even though the latter are often not aways successful because of the common side effects and the damage caused to healthy tissues and organs. Aggressive and metastatic BCs are difficult to treat, thus new studies are needed in order to find new therapies and strategies for managing these diseases. In this review, we intend to give an overview of studies in this field, presenting the data from the literature concerning the classification of BCs and the drugs used in therapy for the treatment of BCs, along with drugs in clinical studies.

## 1. Introduction

BC is considered to be the second most common cancer subtype investigated worldwide [1] and it is estimated that 2.3 million new cases are diagnosed each year worldwide [2]. As per the 2021 WHO statistics, BC has become the most prevalent cancer in the world, and it is predicted that it could constitute more than 30% of all cancers diagnosed in women [3]. Its patterns and trends vary in distinct countries, being the second driving cause of cancer death among women overall—after lung cancer—and the head cause of cancer death in Africa and America among Black and Hispanic women [4,5]. About 5–10% of all BC patients are genetically predisposed to cancers [6,7] and almost 80% of patients with BC are individuals aged >50. Nevertheless, BC is the most common cancer diagnosed and the second most common cause of cancer death in women aged less than 40 years [8]. The number of risk factors of BC is significant and includes modifiable and non-modifiable factors, such as excessive body weight, physical inactivity, and alcohol intake [9]. A recent study suggested that negative emotions also significantly increase the incidence risk for BC [10]. Prevention through mammography screening is fundamental for allowing early therapy and a substantial reduction in BC mortality [11]. BC treatment is multidisciplinary; indeed, breast-conserving surgery with radiotherapy or mastectomy is generally used in early-stage BCs and sentinel node biopsy is used for axillary staging. In most cases, endocrine therapy and/or chemotherapy are needed together with adjuvant and neo-adjuvant systemic therapies that are often used in the majority of women based on proven survival outcomes [12]. The survival rate from BC has improved in recent years; for instance, the 5-year survival rate for non-invasive BC is 99%, while the 5-year survival rate is 85% for invasive BC diffused to the regional lymph nodes. However, the survival rate reduces to 27% if distant parts of the body are also affected [13]. Approximately 70% of BC metastases occur in the bone [14] and metastatic breast cancer (MBC) is currently considered incurable. Lei et al. [15] recently assessed that the incidence and mortality rates of BC increased rapidly in China and South Korea but decreased in the USA. In 2020, the diagnosis and treatment of cancer was negatively affected by the coronavirus disease 2019 (COVID-19) pandemic [16]. Indeed, the reduced access to care due to the closure of healthcare settings along with the fear of COVID-19 exposure have led to delays in diagnosis and treatment bringing to a short-term drop in cancer incidence, followed by an increase in advanced-stage disease and, ultimately, in mortality [17]. Obviously, patients suffering from BC and contemporaneously infected with SARS-CoV-2 have a high probability of mortality [18].

## 2. Classification of BCs

Different classifications have been proposed for BC (Figure 1). It is generally categorized into five major subtypes based on the presence or absence of receptors expressed by tumor cells: luminal A (LumA), luminal B (LumB), HER2-overexpressing (or HER2-enriched or HER2^+^), triple negative breast cancer (TNBC), and basal-like and normal-like (or normal breast tissue-like) [19]. The luminal A, B, and HER2^+^ subtypes are positive for hormone receptors (HRs), i.e., estrogen receptor (ER) and/or progesterone receptor (PR). In luminal B and HER2^+^ subtypes, HER2 overexpression is also noticed. The majority (60–70%) of breast tumors are represented by luminal A and B breast cancers subtypes. The HER2^+^ breast cancer subtype that constitutes 10–15% of invasive breast cancers is characterized by the amplification/activation of the HER2 gene, which results in the HER2 receptor overexpression on the surface of breast cancer cells. TNBC was so named (“triple negative”) because it lacks the expression of the three molecular markers (ER, PR, and HER2), and is the most aggressive subtype, poorly prognosed, often observed in young women, representing the 15–20% of all BCs.

Another hormone receptor (HR) that is not generally classified in HRs is the prolactin (PRL) receptor (PRLR) that is overexpressed in most of the ER^+^ breast tumors. The role of PRLR in the etiology and proliferation of breast carcinoma induced by PRL has been well established [20]. Moreover, another subtype of BC has been described, namely the “claudin-low”, defined by gene expression characteristics—most prominently a low expression of cell–cell adhesion genes, high expression of epithelial–mesenchymal transition (EMT) genes, and stem cell-like/less differentiated gene expression patterns [21,22]. However, a vast degree of heterogeneity has been observed in this subtype of BC, suggesting a need for more investigations into this form of cancer [23]. Recently, HER2 has been named ERBB2 or ErbB2, being part of the ErbB family of receptor tyrosine kinases (also known as epidermal growth factor receptor (EGFR)-family) and consisting of four closely related receptors which includes EGFR (ErbB1/HER1), ErbB2 (HER2/neu), ErbB3 (HER3) and ErbB4 (HER4) [24]. A more recent classification of BCs is based on the presence or absence of molecular markers for ERs or PRs and human epidermal growth factor 2 (ERBB2; formerly HER2): hormone receptor positive/ERBB2 negative (70% of patients), ERBB2 positive (15–20%), and triple-negative (tumors lacking all three standard molecular markers; 15%) [25]. All the subtypes have distinct risk profiles and treatment strategies, depending on tumor subtype, anatomic cancer stage, and patient preferences. BC is one of the first malignancies for which targeted therapies have been used successfully [26] and, in the period from 2010 to 2020, 30 drugs have been approved by the FDA for the treatment of BC [27]. However, side effects, often related to the lack of treatment adherence, are still reported [28]. Despite the efforts made in treating cancers during the past decades, resistance to known chemotherapeutic agents and/or new targeted drugs continues to be a great dilemma in cancer therapy [29]. Drug resistance may be intrinsic—i.e., existing before treatment—or acquired, generally generated after the therapy, but both of them are responsible for most relapses of cancer, one of the main causes of death from BC [30]. Thus, finding new strategies for curbing BC is nowadays a field of great interest. For instance, nanoparticles (NPs) and engineered NPs have been proposed as a next-generation platform with preventive and therapeutic effects against BC [31]. In this review, we highlight the most common therapeutic strategies used to restrain BC and most recent studies for novel compounds designed as potential anticancer drugs for fighting BC in the future.

## 3. Genetic Mutations in BCs

Genetic mutations in specific genes enhance the probability of acquiring breast cancer and may influences its severity. It is known that mutations in the *p53*, *BRCA1*, and *PTEN* genes account for about 10% of familial breast and ovarian cancer cases overall [32]. BReast CAncer genes (*BRCA*) 1 and 2 encode DNA repair enzymes and took their name just because of their association with BC [33,34]. BRCA mutations are highly associated with TNBC, as well as the loss of *BRCA1* or *BRCA2* activity [35]. Phosphatase and Tensin homolog deleted on chromosome 10 (*PTEN*) is a tumor suppressor gene located in the 10q23 region of chromosome 10 that has been found to be mutated in many types of cancers, including BC, and encodes for a dual lipid and protein phosphatase [36]. PTEN inactivity is associated with an aggressive triple-negative phenotype cancer, as in TNBC [37].

## 4. Breast Cancer Therapies

The approach to BC therapy is multidisciplinary and the current major treatments for are represented by surgery, cytotoxic chemotherapy, targeted therapy, radiotherapy, endocrine therapy, and immuno-therapy [38,39]. The plurality of women with early-stage BC is subjected to breast-conserving surgery with radiotherapy or, alternatively, mastectomy and the sentinel node biopsy is utilized for axillary staging, minimizing the demand for axillary dissection in sentinel node-positive women. Neoadjuvant and adjuvant therapies are often used to treat BC and may include chemo- and radiotherapy, hormone and/or immune-therapy and targeted therapy. The major difference is that neoadjuvant (or preoperative) therapies are delivered before the cancerous tumor is surgically removed, helping to reduce the size of the tumor or kill cancer cells that may have spread, whereas the adjuvant therapy is usually used after a cancerous tumor has been surgically removed in order to destroy the remaining cancer cells (Table 1) [12].

Moreover, new methods—such as targeted and immuno-therapy—have been developed to improve patient survival and prognosis. The era of targeted therapies has offered a new avenue in the search for potentially more efficacious strategies. Recently, cancer stem cell (CSC)-targeted therapies, including those inducing CSC apoptosis and inhibiting the self-renewal and division, have been under study [40]. Recently, sirtuins (SIRT1–7) have demonstrated a great potential as biomarkers and/or targets for the treatment of BCs and the dual role of SIRT1–7 as tumor promoters or suppressors in BCs have been widely discussed [41]. Common therapies will be discussed below.

### 4.1. Endocrine Therapy

Endocrine therapy can be used for all patients affected by BC that express ERs and PRs—specifically, LumA, LumB and normal-like BC and as adjuvant and neoadjuvant in several types of BCs [42,43]. However, due to the well-known limits of this type of therapy, the search for new ER inhibitors is still ongoing [44]. The majority (80%) of BC patients are hormone-dependent or ER^+^ [45]. Amongst ER receptors, ERα subtype is the major responsible for estrogen effect in the breast, whereas the activity of ERβ prevents tumor formation in response to estrogens. Recent studies have focused on *ESR1*, which is the gene spanning q24–q27 of chromosome 6 and encodes for ERα, the nuclear transcription factor most commonly implicated in BC [46]. However, when *BRCA1* or *BRCA2* mutations are present, patients have poor survival outcomes and hence screening for *BRCA* mutations might help in strategizing the treatment and improving the survival [47]. For MBC patients with ER^+^ disease, endocrine therapy—with or without the addition of targeted agents—is recommended as first line systemic anti-cancer therapy (SACT) [48,49]. The most commonly used endocrine therapies are based on the use of selective estrogen-receptor modulators (SERMs), selective estrogen receptor downregulators or degraders (SERDs), aromatase inhibitors (AIs), gonadotropin-releasing hormone (GnRH) agonists. However, mutations on the *ESR1* gene may produce resistance to endocrine therapies and patients with *ESR1* mutant–positive metastases are resistant to standard-of-care endocrine therapy and evidence a worst overall survival [50]. However, resistance to endocrine treatment often occurs in BC patients, as assessed by REVERT clinical study. Recently, the combined use of the drug eribulin has been suggested, as it may sensitize the tumor to the hormonal treatment due to the switch of cancer cell phenotype [51].

#### 4.1.1. Selective Estrogen-Receptor Modulators (SERMs)

SERMs specifically control the ERs and limit the progression of breast malignancy regulating particularly ERα, which is mainly responsible for the initiation and progression of BC. They act as antagonists of the transcription process in the BC cell and as agonists in other tissues (bone and endometrium); however, the long-term use of traditionally marketed SERMs is associated with several side effects, such as the development of endometrial cancer and other disorders [52]. SERMs include tamoxifen, raloxifene, bazedoxifene, and lasofoxifene. For over 30 years, tamoxifen—a triphenylethylene SERM (Table 2)—has been the drug of choice solely to treat ER^+^ BC patients. It is a partial nonsteroidal estrogen agonist, which represents the prototype of SERMs, acting as a type II competitive inhibitor of estradiol used for early and metastatic BC. Tamoxifen is metabolized to the more active metabolites 4-OH-tamoxifen and endoxifen by the CYP2D6 and CYP3A4/5 enzymes; thus, may be considered a prodrug [53]. Raloxifene hydrochloride (Evista^®^), a benzothiophene derivative acting as estrogen antagonist was the first SERM approved by the US FDA as a protective and therapeutic agent for postmenopausal osteoporosis. Later, raloxifene was also approved for reducing breast cancer risk, and it has been more recently investigated for breast cancer management [54]. Unfortunately, raloxifene has a reduced bioavailability (not higher than 2%), being subjected to dramatic first pass metabolism due to its poor water solubility; thus, more recently, raloxifene-loaded semisolid self-nanoemulsifying system (SSNES) have been proposed, with minimized globule size in order to improve drug solubility, tumor penetration, and antitumor activity [55]. Lasofoxifene (Fablyn^®^), a SERM with benefits on bone health and potential for breast cancer prevention, has been investigated in mouse models of endocrine therapy-resistant BC with ERα mutations, Y537S and D538G, which have low sensitivity to fulvestrant inhibition, demonstrating its potential as an effective therapy for women with advanced or metastatic ER+ BC expressing the most common constitutively active ERα mutations [56]. Lasofoxifene demonstrated interesting activity in the clinical trial ELAINE 1 in which it was compared with fulvestrant in *ESR1*-mutated MBC patients with progression on CDK4/6i, and all clinical outcomes were in favored lasofoxifene. The association of lasofoxifene with abemaciclib, in phase II trial ELAINE 2, showed efficacy in heavily pretreated patients with *ESR1*-mutated MBC post-CDK4/6i (ASCO 2022) [57]. Bazedoxifene (Duavive^®^) is a synthetic SERM, which received approval by the United States Food and Drug Administration for the treatment of osteoporosis in postmenopausal women. It is now being studied as a repositioned drug for its anticancer activity in different types of cancers including BC [58]. Recently, the combination of BZA and palbociclib showed clinical efficacy and an acceptable safety profile in a heavily pretreated patient population with advanced HR^+^/HER2^−^ BC [59]. However, complicacies due to SERMs remain because of the onset of many dangerous adverse effects as endometrial carcinoma, hot flashes, and VTE (venous thromboembolism), thus novel candidates with no or lower adverse effects for ER+ BC prevention are needed [60].

#### 4.1.2. Selective Estrogen Receptor Downregulators or Degraders (SERDs)

SERDs include fulvestrant and investigational SERDs (AZD9496, AZD9833, LY3484356, GDC-0810, GDC-0927, GDC-9545, and SAR439859), which inhibit ER-mediated cellular proliferation through ERα degradation [61]. Fulvestrant (Faslodex^®^) is a first-generation SERD approved by the FDA in 2007 for the treatment of metastatic luminal BC in postmenopausal patients following progression on prior endocrine therapy with aromatase inhibitors or tamoxifen [62]. It has also shown interesting activity in patients with *ESR1* mutations in the second line treatment setting. It is the unique SERD approved for use as a second-line endocrine therapy effective in treating endocrine therapy-resistant tumors [63]. It can degrade and completely antagonize ERα in all settings tested and has proven to be superior to the aromatase inhibitor anastrozole in clinical trials [64]; it is also used in MBC. Nevertheless, its effectiveness in metastatic disease has been validated, the challenges of drug resistance remain even for this drug [65] and it has the limit of being an injectable with considerable pharmaceutical liabilities [66]. Thus, several SERDs available as oral formulations are under study as monotherapy or in combination with other drugs [67,68,69,70]. GDC-9545 (giredestrant) was identified as an oral SERD with an exceptional preclinical profile [71]. It is now in phase III clinical trials as monotheraphy [72,73], and in association with palcociclib [74,75,76,77] and its use was suggested in order to overcome acquired resistance, since it was observed that BCs with mutant ERα remain sensitive to giredestrant and reversed progesterone hypersensitivity driven by this mutation [78]. Rintodestrant (G1T48) is an orally bioavailable, nonsteroidal SERD that was developed through structure-guided investigations driven by activity in BC cell lines. It was obtained by a structural modification of raloxifene, with the introduction of an acrylic acid side chain [79] and is currently under clinical development for the treatment of patients with HR^+^ breast cancer [80,81,82]. Amcenestrant (SAR439859) is a nonsteroidal, orally bioavailable, SERD studied in clinical trials (AMEERA) in postmenopausal women with HR^+^/HER2^−^ advanced breast cancer [83] and in association with CDK4/6 inhibitors, such as palbociclib for previously untreated ER^+^/HER2^−^ advanced breast cancer [84]. AZD9833 (camizestrant) is an oral tricyclic indazole that has demonstrated high potency in preclinical cancer models as SERD and pure ER antagonist [85]. Two phase III trials are ongoing to evaluate the effectiveness of camizestrant in combination with palbociclib: SERENA-4 (NCT04711252) is a randomized, multicenter, double-blind, trial for evaluating the safety and efficacy for patients with ER^+^/HER2^−^ advanced breast cancer who have not received systemic treatment in the advanced disease setting [86]; SERENA-6 is a study to assess the efficacy and safety of camizestrant in combination with palbociclib or abemaciclib in patients with ER^+^/HER2^−^ MBC with detectable *ESR1* mutations who have not experienced disease progression on first-line therapy, compared with aromatase inhibitors [87]. Elacestrant (RAD1901) is the first oral selective ER degrader demonstrating a significant progression-free survival (PFS) improvement compared to the standard-of-care (SOC) endocrine monotherapy both in the overall population and in patients with *ESR1* mutations, with manageable safety in a phase III trial (EMERALD) for patients with ER^+^/HER2^−^ advanced BC or MBC [88]. Imlunestrant (LY3484356) is a novel orally bioavailable SERD with pure antagonistic properties that showed a favorable profile as monotherapy in the study EMBER (NCT04188548) in ER^+^, HER2^−^ advanced breast cancer [89]. AZD9496, an oral nonsteroidal small molecule inhibitor of ERα, is a potent and selective antagonist and downregulator of ERα in vitro and in vivo in ER^+^ models of breast cancer [90,91]. It is well tolerated with an acceptable safety profile, showing evidence of prolonged disease stabilization in heavily pretreated patients with ER^+^/HER2^−^ advanced breast cancer [92]. Other oral investigational drugs for BC have been summarized by Chen et al. (2022) [93].

#### 4.1.3. Aromatase Inhibitors (AIs)

AIs—such as anastrozole, letrozole, and exemestane—are effective targeted therapy in patients with ER^+^ BC, used in early and metastatic BC and in pre- and post-menopausal BC patients. They act as inhibitors of the CYP19 aromatase enzyme that catalyzes essential steps in estrogen biosynthesis, blocking the production of estrogens and, thereby, the downstream ER signaling. Since AIs deplete systemic estrogen levels in post-menopausal patients by blocking the conversion of androgens to estrogens, they are more effective than SERMs because they block both the genomic and nongenomic activities of ER. To date, first-generation (e.g., aminoglutethimide), second-generation (e.g., formestane and fadrazole), and third-generation (e.g., anastrozole, letrozole, and exemestane) AIs have been approved by the FDA and the third-generation is currently used in the standard treatment for postmenopausal breast cancer [94]. AIs can be categorized into Type I (steroidal) and Type II (non-steroidal) inhibitors, based upon their structure. Type I inhibitors, known as aromatase inactivators, have a steroidal structure similar to androgens and inactivate the enzyme irreversibly by blocking the substrate-binding site; whereas type II inhibitors are nonsteroidal and their action is reversible [95]. Most of the type II inhibitors are heterocyclic compounds—such as azoles, chromene, coumarin, xanthone, triphenylethylene, indole, sulfonamide, pyrimidines, pyridine, quinolone, and thiourea [96]. The treatment with AIs in early BC patients determined improvements of clinical psychological features, and health related quality of life was observed in BC survivors [97]. However, the therapy with AIs is associated to several side effects related to estrogen depletion in the body, which include menopausal symptoms—such as hot flashes, insomnia, accelerated bone loss—leading to higher osteoporosis risk, and most significantly, arthralgias [98]. Moreover, the potential role of AIs in the onset of autoimmune disorders—such as rheumatoid arthritis [99], anti-synthetase syndrome [100], and anti-phospholipid syndrome [101]—has been described. Finally, postmenopausal women with BC treated with AIs have an increased risk of cardiovascular events in comparison with tamoxifen [102].

#### 4.1.4. Gonadotropin-Releasing Hormone (GnRH) Agonists

GnRH (also known as luteinizing hormone-releasing hormone (LHRH), gonadoliberin, luliberin, gonadorelin) is produced by the hypothalamus in the form of decapeptide. It binds specifically the GnRH receptor (GnRH-R) on the surface of gonadotropic cells in the anterior pituitary gland and promotes the production and the release of gonadotropins in both adult men and women. GnRH-R is a G-protein coupled receptor, which activates the mitogen-activated protein kinase (MAPK) cascade leading to growth and proliferation of the cells. GnRH-R is overexpressed in cancers that are dependent on gonadal steroids such as breast cancer, representing the 50% of the cases [103]. GnRH agonists bind GnRH-R in the pituitary gland, resulting in the secretion and initial surge of FSH and LH, thus stimulating the production of serum testosterone or estrogen. Thereafter, the negative feedback at the pituitary gland causes the downregulation of GnRH receptors suppressing the downstream effects of follicle-stimulating hormone and luteinizing hormone, ultimately leading to a decreased estrogen production in premenopausal ovaries [104]. GnRH agonists cause ovarian suppression that can aid fertility preservation useful for the successive fertility treatments and pregnancies in women with a previous diagnosis of BC [105]. The most common GnRH agonists are goserelin, triptorelin, buserelin, and leuprolide [106,107], used only in premenopausal women with BC. GnRH agonists are often administered in combination with cyclofosfamide [108], tamoxifen [109], or cyclin-dependent kinase 4/6 inhibitor [110,111] in the adjuvant or metastatic settings in patients with premenopausal or perimenopause endocrine positive BC. One advantage of the use of GnRH agonists is the protection of ovarian reserve that was observed with the use of goserelin during chemotherapy, even though the exact mechanism has not been fully elucidated yet [112]. The impact of GnRH agonists on ischemic heart disease has not been established, but some authors report an increased risk of cardiovascular toxicity after treatment with GnRH agonists, especially in men with prostate cancer [113,114], whereas it has not been demonstrated in women with BC. A recent study found that female patients with breast cancer receiving GnRH agonists had a lower risk of developing ischemic heart disease than patients not receiving them [115].

#### 4.1.5. Complete Estrogen Receptor Antagonists (CERANs)

CERANs are small molecules that degrade ER as a SERD, but also hamper ER function, particularly blocking the activity of both the AF1 and AF2 transcriptional activation functions, inhibit ER-driven breast cancer cell growth, and induce ER degradation [116]. OP-1250 is an orally bioavailable CERAN currently in phase I/II open-label, multi-center, dose-escalation and expansion clinical trial (NCT04505826) for the treatment of ER^+^ advanced BC and/or MBC [117,118]. OP-1250 potently competes with the endogenous activating estrogenic ligand 17-beta estradiol for binding the ligand binding pocket [119].

#### 4.1.6. Selective Estrogen Receptor Covalent Antagonists (SERCAs)

Several years ago, covalent targeting of ERα had been accomplished using estrogenic (ketonoestrol aziridine) and antiestrogenic (tamoxifen aziridine) affinity labels as tools to map the hormone-binding domain [120]. SERCAs are addressed specifically to a positionally non-conserved cysteine 530 (C530) in the ligand-binding pocket of ERα that could be covalently modified [121,122]. H3B-5942 is the first-in-class experimental oral compound showing exquisite selectivity for C530 and antitumor activity superior to fulvestrant therapy in BC xenograft models with ESR1 WT or ESR1 Y37S mutation [123]. It is known for being the precursor of compound H3B-6545, which was suggested for the potential treatment of endocrine therapy-resistant ERα+ BC harboring wild-type or mutant ESR1 (clinical trials: NCT03250676, NCT04568902, NCT04288089) [124]. It is noteworthy that the name SERCA must not be confused with sarcoplasmic reticulum (ER/SR) Ca^2+^-ATPase, which possesses the same acronym and is also involved in studies regarding BC [125,126,127].

#### 4.1.7. Selective Human Estrogen Receptor Partial Agonist (ShERPA)

ShERPAs target and bind ER into the nucleus, causing ER translocation to extra-nuclear sites, which cause ER^+^ tumor cell growth inhibition [128]. They are based on the benzothiophene scaffold of raloxifene and mimics the effects of estradiol [129]. Two ShERPAs, namely BMI-135 and TTC-352, have been studied [130] and TTC-352 has completed successfully the phase I clinical trial for treatment of hormone-refractory HR^+^ BC demonstrating a favorable safety profile and early clinical evidence of antitumor activity [131].

### 4.2. Chemotherapy

Historical chemotherapic cancer treatments include direct DNA-damaging chemotherapy, such as platinum salts (cisplatin), alkylating agents (cyclophosphamide), antimetabolites (methotrexate, pemetrexed, gemcitabine, capecitabine, 5-fluorouracil), and topoisomerase I/II inhibitors. Chemotherapy represents the first line treatment option for BC with no hormone receptors [132]. For HR^+^/HER2^−^ MBC, international guidelines recommend endocrine therapy as first-line treatment, except in case of ‘visceral crisis’, for which the cytotoxic chemotherapy is preferable [133]. Moreover, as neoadjuvant and adjuvant therapy, chemotherapy can be used also in other types of BC, even though its role is still controversial, especially in certain cases [134,135]. Unfortunately, patients frequently develop resistance. Chemotherapy is often used in association with immune checkpoint inhibitors to treat BC.

#### 4.2.1. Targeted Therapy

In 1971, Judah Folkman first proposed that tumor angiogenesis could serve as a potential target for anticancer therapy [136]. Since then, targeted therapies against vulnerabilities in the key cell signaling pathways within the cancer cell began to be developed in the 1990s. Important advances have been obtained with targeted therapies in all the forms of cancer. Specifically, targeted therapies are represented by drugs that target particular enzymes, growth factor receptors, and signal transducers, thus interfering with various oncogenic cellular processes [137]. HER2, encoded by the oncogene ErbB2, is the transmembrane protein in human cells [138] that is involved in regulating cell proliferation, differentiation, and apoptosis through the activation of signal transduction by homo- or hetero-dimerization [139]. It was among the first to be discovered [140] and now represents one of the most promising targets in oncology research, especially in BC; numerous therapeutic agents are addressed to this target [141,142].

##### Topoisomerase (TOP) Inhibitors

DNA topoisomerases (TOPs) are crucial enzymes that modulate the double helix of DNA, by regulating DNA under- and overwinding and the removal of tangles and knots from the genome. TOP (or TOPO) inhibitors cause DNA double-strand breaks during DNA replication: they have been widely used as anti-cancer drugs during the past 20 years [143]. TOPO1 (TOP1 or TOPOI) inhibitors selectively trap TOPO1 cleavage complexes resulting in DNA double-strand breaks during replication, which are repaired by homologous recombination. Camptothecin (Table 3) is a natural alkaloid from the Chinese tree *Camptotheca acuminata* acting as TOPO1 inhibitor. Due to its extremely low solubility, the original natural product is not used as a drug but the semisynthetic derivatives topotecan and irinotecan were employed and clinically approved as TOPO1 inhibitors [144]. Camptothecin and its derivatives are currently used as second- or third-line treatment for patients with endocrine-resistant BCs [145,146]. Etirinotecan pegol (NKTR-102) is a long-acting topoisomerase-I inhibitor used for patients with MBC and brain metastases. However, in the phase III ATTAIN randomized clinical trial, no statistically significant difference in survival outcomes was observed in patients treated with etirinotecan pegol and patients treated with chemotherapy of physician’s choice [147]. These findings are questionable [148] and further studies are needed.

Type II topoisomerases (TOP2s, TOPO2s, or TOPOIIs) alter DNA topology through the generation of a transient double-stranded break in one segment of DNA, thus allowing another segment to pass through the DNA gate. Adriamycin, also known as doxorubicin, is a DNA TOPOII inhibitor belonging to the family of anthracycline anticancer drugs isolated from *Streptomyces peucetius* [149]. It is currently one of the most effective chemotherapeutic drugs for BC patients. However, several studies showed that adriamycin chemotherapy can cause adverse cardiac reactions in BC patients [150], and even heart failure [151]. Moreover, growing chemotherapeutic resistance to adriamycin has emerged as one of the essential reasons for treatment failure and poor outcome in BC [152,153]. Finally, it has been recently suggested that adriamycin therapy may accelerate BC metastasis in a SIRT7/TEK (TIE2) dependent manner [154]. Several mechanisms have been proposed to explain adriamycin resistance [155], including circular RNA (circRNA) in TNBC [156], RBM38 that may be negatively regulated by miR-320b, accelerating drug resistance in BC [157] and several strategies are addressed to overcome adriamycin resistance. Placenta-specific 8 (also known as PLAC8 or Onzin) is a highly conserved protein functioning as an oncogene or tumor suppressor in various tumors: it was suggested as a therapeutic target, through the participation of p62, in BC treatment and for its potential clinical application in overcoming adriamycin resistance [158]. Other promising strategies for adriamycin-resistant BCs include targeting exosomes or exosomal miR-222 [159]. The most common therapies for BC in association are: adriamycin/cyclophosphamide (AC), adriamycin/cyclophosphamide/paclitaxel (AC-T), cyclophosphamide/methotrexate/5-fluorouracil (CMF), and docetaxel/cyclophosphamide (TC) (Waks 2019). Neoadjuvant chemotherapy aids in debulking the tumor before surgery and is recommended for locally advanced BC patients [160]. Adjuvant chemotherapy aids in eradication of remaining BC tumor cells as well as undiscovered micrometastases and was demonstrated to reduce mortality by one third [161].

##### Drugs Targeting Tubulin

Taxanes, commonly used to treat different types of cancer, including BC, inhibit mytosis by binding the β-tubulin subunit of microtubules and induce apoptosis by preventing depolymerization of microtubules. Although they are effective against many cancer types, usual adverse effects are arrhyhtmias, hypersensitivity reactions, myelosuppression, nausea, vomiting, and neuropathy [162]. The broad-spectrum anticancer drug paclitaxel is currently one of the most used taxanes, besides other formulations such as docetaxel, cabazitaxel, and nab-paclitaxel. Paclitaxel is a diterpene pseudoalkaloid with a taxane ring as nucleus (C_47_H_51_NO_14_), first extracted and isolated from the bark of the Pacific yew tree *Taxus brevifolia* Nutt. Then, it was obtained with semi-synthetical procedures, starting from a paclitaxel precursor from the European yew tree’s needles. Its interaction with microtubules impairs the abnormal proliferation and mitosis rate of tumor cells. Specifically, paclitaxel enters in the microtubule lattice through small openings and binds β-tubulin, interfering with the dynamic process [163], leading to a strengthening between the tubulin subunit’s lateral contacts. Therefore, the depolymerization is suppressed and the microtubules are stabilized [164]. The application of paclitaxel is difficult because of its hydrophobic properties; thus, fat-soluble solvents such as cremophor are applied for injection purposes in order to improve its poor solubility. However, solvent-based paclitaxel can cause side reactions, including bone marrow suppression, allergic reactions, peripheral neuropathy [165], and cosolvent-induced toxicity. In order to improve the water solubility and safety of paclitaxel, nab-paclitaxel (nanoparticle albumin-bound paclitaxel)—a 130 nm particle solvent-free formulation, comprising albumin nanoparticles and paclitaxel with non-covalent bonds—has been successfully used for microtubule depolymerization inhibition [166]. Food and Drug Administration (FDA) lists nab-paclitaxel as a vital drug for the treatment of non-small cell lung cancer, pancreatic, and breast cancers, and it has recently suggested that nab-paclitaxel may promote the cancer-immunity cycle, acting as a potential immunomodulator [167]. Docetaxel, a semi-synthetic and side-chain analog of taxol, produced from 10-deacetylbaccatin III [168] has been used to treat MBC in combination with capecitabine after failure of first-line anthracycline-based treatment or in combination with HER2-targeted therapy in HER2^+^ MBC, or as monotherapy in second or later palliative lines [169]. Adverse effects are represented by of neutropenia, febrile neutropenia, neuropathy, onycholysis, epiphora, and toxicity, often leading to treatment discontinuation. Moreover, docetaxel is a highly lipophilic drug with very poor water solubility and instability [170]. Cabazitaxel is a recent taxoid agent showing in vitro and in vivo activity against cell lines and tumors resistant to docetaxel and paclitaxel [171]. Promising results were shown for the use of cabazitaxel in MBC patients previously treated with taxanes, and specifically after the development of taxane resistance [172].

##### Drugs Targeting Actin

Metastasis is a multi-step process driven by the dynamic reorganization of the actin-myosin cytoskeleton [173]. About five decades ago, it was found that actin filaments were disrupted in the malignant transformed cells. Indeed, migration and establishment of metastatic colonies require dynamic cytoskeletal modifications, characterized by polymerization and depolymerization of actin. Thus, actin became a possible and useful therapeutic target for chemotherapy and also an indicator for the efficacy of chemotherapeutic drugs [174] and several drugs hampering actin polymerization have been studied. Actin cytoskeletal modifications are initiated and regulated mainly by extracellular matrix (ECM) factors through integrin-focal adhesion kinase (ITG-FAK) signaling. Nimbolide, a terpenoid lactone found in leaves and flowers of *Azadirachta indica*, is widely known as ‘neem’ and in a recent study by Arumugam [175] it was demonstrated to inhibit TNBC growth, by altering AKT/mTOR and MAPK signaling, thereby inducing apoptosis. Nimbolide also inhibits ITG-FAK and Rac1/Cdc42-signaling pathways leading to a subsequent induction of actin depolymerization that has been suggested as a promising approach to reduce metastasis.

##### Trophoblast Cell Surface Antigen 2 (TROP2) Inhibitors

TROP2 is a type I transmembrane glycoprotein and calcium signal transducer first identified as a cell surface marker for trophoblast cells in the placenta [176]. Successively, a large body of work has implicated TROP2 as a major tumorigenic factor, being overexpressed in a number of malignant tumors, including BC and especially TNBC [177], thus representing a valid drug target [178,179]. Moreover, the use of specific monoclonal antibody targeting TROP2, such as TrMab-6 and TrMab-29, has been suggested to evaluate TROP2 expression in BCs [180,181].

#### 4.2.2. HR^+^ Targeted Therapies

##### mTOR Inhibitors

The mTOR pathway was studied in the late 1970s, after the discovery of the antifungal metabolite rapamycin produced by *Streptomyces hygroscopicus*. TOR took its name from rapamycin: “target-of-rapamycin” [182]. mTOR is a Ser/Thr kinase belonging to the phosphoinositide 3-kinase related kinase (PIKK) family and exists as two distinct complexes, mTORC1 and mTORC2. mTOR signaling is often overactive in multiple cancer types including breast cancer [183] and mTOR antagonist are employed in clinical studies for BC, especially in association with cytotoxic chemotherapy. Rapamycin (sirolimus, HY-10219, Rapamune) possesses immunosuppressive and antiproliferative properties and is under study in HR^+^/HER2^−^ BCs, generally in association with tamoxifen. Rapamycin regulates cell growth and survival through the mediation of the EGFR signaling pathway [184]. The other mTOR inhibitors are often mentioned as “mammalian target-of-rapamycin” inhibitors, that may cause confusion, since rapamycin is already an inhibitor. Everolimus (Afinitor, HY-10218) is a derivative of sirolimus that binds with high affinity its intracellular receptor, 12-kD FK506-binding protein (FKBP12), a protein belonging to the immunophilin family, forming a complex that is an allosteric inhibitor of mTORC1. Everolimus is approved for the treatment of ER^+^, HER2^−^ MBCs in combination with hormonal therapies [185] and, considering the common resistance to CDK4/6 inhibitors, it is also used in recurrent BCs treated with CDK4/6 inhibitors in clinical settings [186]. Temsirolimus (CCI-779, HY-50910, Torisel) is an ester, analog of rapamycin, with improved aqueous solubility and pharmacokinetic properties that binds the FKBP12 forming a complex that is studied in combination with neratinib in ER^+^/HER2^+^ MBC [187]. Sapanisertib (TAK-228, MLN0128) is an oral, potent, and highly selective inhibitor of mTOR kinase exhibiting activity against both mTORC1 and mTORC2. It is studied in combination with exemestane or fulvestrant in postmenopausal women with ER^+^/HER2^−^ advanced BC [188] and is currently under study in combination with another mTOR inhibitor, serabelisib (TAK-117), and paclitaxel for gynecological cancers, including BC [189].

##### Phosphatidylinositol 3-Kinase (PI3K) Inhibitors

PI3Ks are lipid kinases that regulate various cellular processes such as proliferation, adhesion, survival, and motility. PI3K phosphorylates phosphatidylinositol-4,5-bisphosphate (PIP2) at the 3′ position on its inositol ring, converting PIP2 to phosphatidylinositol-3,4,5-trisphosphate (PIP3) [190]. Dysregulated PI3K pathway signaling occurs in one-third of human tumors [191]. PI3Ks are classified into three classes (I, II, and III) based on their primary conformations and binding substrates. Class I PI3K is grouped into class IA (PI3K isoforms α, β, and δ) and class IB (PI3K isoform γ) [192]. PI3Kα is the most commonly associated with solid tumors via gene amplification or mutations of the *PIK3CA* gene, which encodes the p110α catalytic subunit of PI3Kα. Approximately 30% of BCs and about 40% of HR^+^/HER2^−^ BC bear aberrant PI3Kα signaling [193]. Several PI3K inhibitors are used as monotherapy or in combination therapies for various cancer, including BC [194]. Alpelisib (NVP-BYL719, Piqray, Table 4) is an oral PI3Kα selective inhibitor that was approved by FDA in 2019 for *PIK3CA*-mutated, HR+/HER2^−^ advanced BC in combination with fulvestrant [195]. Other PI3K inhibitors are in clinical study, such as Inavolisib (GDC-0077), a PI3Kα-specific inhibitor that also promotes degradation of mutant p110α [196]. It has demonstrated encouraging preliminary results in a phase I/Ib clinical study in patients with *PIK3CA*-mutated HR^+^/HER^−^ MBC as a monotherapy, and in combination with palbociclib and/or endocrine therapy. A phase III study of inavolisib + palbociclib + fulvestrant is undergoing (NCT03006172) [197]. Taselisib (Genentech), a dual PI3Kα/PI3Kδ inhibitor, entered phase III studies in BC [198]. However, the modest clinical benefit and considerable side effects (probably related to PI3Kδ inhibition) have led to its discontinuation [199]. Buparlisib (BKM120) and Pictilisib (GDC-0941) are pan-PI3K inhibitors and the second is one of the very first pan-class I selective PI3K inhibitors evaluated in patients with advanced cancer [200], being currently in phase Ib clinical study in combination with paclitaxel for advanced BC [201]. Buparlisib is an oral 2,6-dimorpholino pyrimidine derivative that potently inhibits the PI3K downstream signaling, including the downregulation of p-Akt and p-S6R and apoptosis of cancer cells [202,203]. Recent clinical studies are ongoing for the combined use of buparlisib (or alpelisib) and endocrine therapies [204,205].

##### Protein Kinase B (Akt) Inhibitors

Akt (or PKB) is a Ser/Thr kinase that is phosphorylated after phosphatidylinositol-3,4,5-triphosphate formation, leading to the downstream activation of the mTORC1 and mTORC2 complexes. The Akt kinase family comprises three isoforms: AKT1, AKT2, and AKT3 that are encoded by different genes with high sequence homology and display a conserved protein structure [206]. AKT activity is controlled in an Akt-dependent manner via phosphorylation and dephosphorylation. In cancer cells, AKT1 is implicated in proliferation and growth, promoting tumor initiation and suppressing apoptosis, while AKT2 regulates cytoskeleton dynamics, favoring invasiveness and metastatization. The function of AKT3 hyperactivation is still controversial. However, a potential stimulation of cell proliferation has been hypothesized [207]. According to the current knowledge about isoform-specific effects in BC, the inhibition of a specific isoform alone is not recommended [208]. Akt inhibitors have been classified into three categories depending on their mechanism of action: ATP-competitive inhibitors (capivasertib, ipatasertib, uprosertib, and GSK690693) that reduce the phosphorylation of Akt by competing with ATP; allosteric inhibitors (MK-2206), which prevent Akt from interacting with its substrate by causing conformational transitions in enzymic structure and finally irreversible inhibitors, which are less common. Capivasertib and ipatasertib (GDC-0068) are pan-AKT inhibitors and have been widely tested in phase I and II clinical trials as monotherapy or in association with chemotherapy or endocrine therapy. Then, phase III trials are ongoing in HR^+^ and TNBC [207].

##### PI3K/Akt/mTOR (PAM) Inhibitors

The three protein components—PI3K, Akt, and mTOR complexes—are primary regulators of the PAM pathway, which is associated with cell growth, survival, and proliferation. Targeting this pathway has been largely studied in cancer treatment for endocrine-resistant BCs, given the increasing resistance to endocrine therapy with upregulation of the PAM pathway. Most recent studies were widely summarized in the literature [209]. However, to increase the therapeutic benefit and overcome the resistance to PAM inhibitors, drug combinations are needed to obtain the highest efficacy and lowest toxicity rate [210,211].

#### 4.2.3. HER2-Positive (HER2^+^) Targeted Therapies

HER2 is a membrane tyrosine kinase which gene is overexpressed or amplified in about 20% of BCs, playing a critical role in cellular transformation and carcinogenesis. Before the occurrence of targeted therapy, patients with HER2^+^ BC had an increased risk of recurrence and death, but since then their outcomes have substantially improved and the situation has changed considerably [212]. Blocking the HER2 pathway is considered a good therapy for BC and at present, several HER2-targeted agents are available to treat HER2-overexpressing BC [213].

##### Tyrosine Kinase Inhibitors (TKIs)

Lapatinib is a dual inhibitor of HER1 and HER2 receptors that competitively and reversibly binds their intracellular ATP-binding domains and slows tumor growth. It received its first FDA approval in 2007 and has been used in combination with capecitabine for previously treated MBC overexpressing HER-2 [214]. Since 2010, lapatinib has been approved in association with letrozole in the treatment of postmenopausal women with advanced HER2^−^ and HR+ BC [215]. Recently, the association of lapatinib with trastuzumab as neoadjuvant therapy for HER2^+^ early BC has shown a better outcome over monotherapy with trastuzumab (CHER-Lob trial) [216].

Neratinib (HKI-272, Nerlynx™) is an oral, irreversible inhibitor of HER1, HER2, and HER4 receptors used in the treatment of BC [217]. In 2017, it received its first approval for HER2^+^ early-stage BC in the US [218]. Pyrotinib is an oral, irreversible pan-ErbB (or pan-HER) kinase inhibitor with activity against HER1, HER2, and HER4 [219]. The combination of PYR and ADM has shown synergistic effects both in vitro and in vivo. It has been demonstrated that PYR suppresses the proliferation, migration, and invasion of breast cancers through the downregulation of the Akt/p65/FOXC1 pathway [220]. The most recent TKI studied is tucatinib (Tukysa) that received its first approval by the FDA in 2020 to treat HER2^+^ MBC [221]. Compared to the other TKIs, tucatinib is highly selective and is more than 1000-fold specific for HER2 than EGFR. Moreover, it is better able to penetrate central nervous system (CNS) than lapatinib and neratinib, thus it can a very interesting potential treatment for MBC with CNS metastases [222].

##### Monoclonal Antibodies

Monoclonal antibodies are an effective therapeutic strategy for HER2^+^ BC and have been around for more than 20 years. Trastuzumab, a recombinant antibody targeting HER2, was the first biological drug approved by the FDA for the treatment of HER2^+^ BC in 1998. Trastuzumab is a first-line treatment option for early and advanced HER2^+^ BC, in mono- or combined therapy, due to its established safety and profile of efficacy. It is used in intravenous and subcutaneous formulations which have similar efficacy and safety. However, the main challenge of trastuzumab-based treatment is represented by the onset of resistance phenomena.

Pertuzumab is a recombinant, humanized monoclonal antibody that targets the extracellular dimerization domain of HER2 (domain II), distinct from the binding site of trastuzumab (domain IV) [223]. It is approved by FDA for use in association with trastuzumab and docetaxel in MBC, and in combination with trastuzumab and chemotherapy as neoadjuvant or adjuvant therapy in non-metastatic disease. The combination of pertuzumab and trastuzumab has shown low cardiac toxicity and favorable efficacy outcomes as demonstrated by the BERENICE study [224].

Margetuximab (MGAH22) is a second-generation monoclonal antibody for the treatment of HER2^+^ breast cancer and other cancers [225] that binds the same epitope on HER2 as trastuzumab. Margetuximab-cmkb (MARGENZA) received approval by the FDA on 16 December 2020, for its usage in combination with chemotherapy for the treatment of for advanced or HER2^+^ MBC [226]. Recently, the combination of margetuximab or T-DXd with small molecule inhibitors such as TKIs has been suggested to further improve the efficacy of monoclonal antibodies in the treatment of HER2^+^ MBC patients [225].

##### Antibody-Drug Conjugates (ADCs)

ADCs derive from the conjugation of traditional chemotherapy agents with monoclonal antibodies. Recent studies are addressed to ADC formed by monoclonal antibodies and inhibitors of TOPOI or TROP2, which are overexpressed in BC. Trastuzumab emtansine (Kadcyla, T-DM1) was the first ADC approved by the FDA on 22 February 2013 for HER2^+^ BC [227]. T-DM1 is composed of a trastuzumab backbone linked via a thioether linker to mertansine (also known as DM1), which is a tubulin polymerization inhibitor. On 20 December 2019, the FDA granted accelerated approval for the treatment of unresectable or metastatic HER2^+^ BC to Fam-trastuzumab deruxtecan-Nxki (T-DxD, Enhertu^®^), composed of trastuzumab and topoisomerase I inhibitor (DXd) [228]. Current studies compare the efficacy of T-DXd and T-DM1 [229]: the randomized phase III trial DESTINY-Breast03 showed a superior progression-free survival and a manageable safety profile for T-Dxd [230]. Moreover, trastuzumab deruxtecan has demonstrated a high intracranial response rate in patients with active brain metastases from HER2^+^ BC in the TUXEDO-1 trial (NCT04752059, EudraCT 2020-000981-41) [231]. Datopotamab deruxtecan (Dato-DXd, DS-1062) is an ADC composed of a humanized anti-TROP2 IgG1 monoclonal antibody attached to a highly potent topoisomerase I inhibitor payload (an exatecan derivative DXd) via a stable cleavable linker. It is now in phase III clinical trial for the treatment of metastatic TNBC (TROPION-Breast02, NCT05374512) [232].

#### 4.2.4. HER2-Negative (HER2^−^) Targeted Therapies

##### Poly(ADP-Ribose) Polymerase Inhibitors (PARPi)

PARPs represent a family of 17 proteins that are activated by DNA damage. *BRCA1* and *BRCA2* are tumor-suppressor genes that encode proteins involved in the repair of DNA double-strand breaks. Cells that lack functional *BRCA1* or *BRCA2* are sensitive to PARP inhibition; thus, PARPi are useful agents in the treatment of BC patients with germline mutations in DNA repair genes, particularly those with deleterious *BRCA1* and *BRCA2* mutations, which constitutes 3–4% of all women with breast cancer and includes 10–20% of those with TNBC [233]. Olaparib (AZD-2281, Lynparza^™^) is an oral, small molecule, it is a PARP1/2 inhibitor first approved in 2014 for *BRCA* mutation-positive ovarian cancer [234]. On 8 January 2018, the US FDA granted approval of olaparib for patients with germline *BRCA* mutations (g*BRCA*m) HER2^−^ MBC [235] and on October 2018, the PARPi talazoparib (BMN-673, Talzenna) received US FDA approval [236]. Veliparib (ABT-888) is generally used as neoadjuvant therapy in TNBC, even though its effectiveness is still controversial [237]. It has demonstrated interesting results in the BROCADE3 trial as a maintenance monotherapy following chemotherapy of veliparib in combination with carboplatin/paclitaxel for HER2^−^ advanced germline g*BRCA*m BC [238,239]. The phase III Bravo trial investigating the role of the PARPi niraparib (MK-4827) versus chemotherapy in *BRCA*-mutated BC, was prematurely closed because of a severe discontinuation rate in the control arm [240]. The phase III clinical trial ZEST (NCT04915755) is ongoing to assess the efficacy of niraparib in patients with wild-type or mutated tumor *BRCA* HER2^−^ TNBC who have detectable circulating tumor DNA (ctDNA) after completion of definitive therapy [241]. Pamiparib (BGB-290) has been approved in China for the treatment of ovarian cancer, fallopian tube cancer and peritoneal cancer [242] and phase II studies are evaluating its efficacy as treatment for HER2^−^ BC with *BRCA* mutation (NCT03575065) [243].

##### Cyclin-Dependent Kinase (CDK) 4/6 Inhibitors

CDK4/6 inhibitors belong to a new class of drugs that interrupt proliferation of malignant cells by inhibiting the progression through the cell cycle. Up to now, three different generations of CDKs inhibitors have been developed [244]. The third generation CDKs selectively target CDK4/6. Specifically, palbociclib, ribociclib, and abemaciclib were recently approved in combination with anti-estrogen therapy for the treatment of advanced and/or metastatic HR^+^/HER2^−^ BC patients [245].

##### Antibody-Drug Conjugates (ADCs)

Sacituzumab govitecan (IMMU-132) is a third-generation ADC, composed of anti-human trophoblast cell-surface antigen 2 (Trop-2) monoclonal antibody, a hydrolyzable CL2A linker, and a cytotoxin (SN38), which inhibits topoisomerase 1, which is useful against TNBC cells. Trop-2 is present in BC cells, therefore, the anti-Trop-2 antibody allows IMMU-132 to specifically deliver SN-38 to the BC cells and the surrounding tumor via the cleavable linker [246]. Sacituzumab govitecan (Trodelvy™) was approved by FDA in 2020 as a third-line treatment for metastatic TNBC, in patients with a history of two prior metastatic treatments [247]. Other ADCs under study are widely described by Liu et al. (2022) [248].

#### 4.2.5. Targeted Protein Degradation (TPD) Technologies

Unlike interfering with protein-protein interaction or other classical approaches, TPD triggers the protein degradation pathway. TPD technologies used for BCs include PROteolysis Targeting Chimeras (PROTACs) and LYsosome Targeting Chimeras (LYTACs), which determine ER degradation taking advantages of protein destruction mechanism in cells and compensates the shortcomings of traditional small molecular inhibitors. Ribonuclease-targeting chimeras (RIBOTAC) and small interfering RNA (siRNA) may inhibit the production of ER protein and are the subject of several recent medicinal chemistry studies [249,250].

##### Proteolysis-Targeting Chimeras (PROTACs)

The technology termed PROTAC, proteolysis targeting chimera, has been recently developed for inducing protein degradation by a targeting molecule [251,252]. PROTACs has been labeled with the mantra of being able to “drug the undruggable” [253,254,255]. A PROTAC is a heterobifunctional small molecule with three chemical elements: a ligand, binding a target protein (the protein of interest, POI), a ligand binding E3 ubiquitin ligase, and a linker that conjugates these two ligands. PROTAC is a chemical knockdown strategy that degrades the target protein through the ubiquitin-proteasome system. Unlike the competitive- and occupancy-driven process of traditional inhibitors, PROTACs have a catalytic mode of action, thus they have the potential to degrade the target pathogenic proteins and regulate the related signaling pathways, which cannot be achieved by traditional therapy (inhibitor/activator) [256]. The resulting PROTAC molecule then specifically recruits the E3 ligase ubiquitin machinery to the target protein of interest (POI) and exploits the cellular ubiquitin proteasome system (UPS) or lysosomal degradation pathways to degrade the protein [257]. The first PROTAC, reported by the groups of Craig M. Crews and Raymond J. Deshaies in 2001, showed the degradation induction of methionine aminopeptidase-2 (MetAp-2) [258]. Then, PROTACs were demonstrated to induce degradation of the androgen receptor and ER, hence expanding the target scope [259]. In the literature other names are also used for these compounds, such as specific and non-genetic IAP-dependent protein erasers (SNIPER); degrader; degronimids; PROteolysis TArgeting Peptide (PROTAP); Protein Degradation Probe (PDP). ARV-471 is a novel, potent, orally bioavailable PROTAC that selectively targets the ER and is currently in phase I/II clinical trial in association with palcociclib (NCT04072952) and as monotherapy in the VERITAC phase II for the treatment of ER+/HER2^−^ locally advanced or metastatic BC [260,261].

##### Lysosome Targeting Chimeras (LYTACs)

The lysosomes represent another protein degradation system within the cell. LYTACs overcome the limitations of PROTACs and other degrader systems, which cannot degrade extracellular proteins and may have greater potential, since they are bifunctional molecules that bind both extracellular proteins and membrane-bound proteins, which are the products of approximately 40% of all protein-encoding genes [262]. PROTAC and LYTAC in combination could reduce the extracellular membrane and intracellular EGFR protein levels in BC cells: this association has been suggested as a potential strategy for treating EGFR antagonist resistance [263].

##### Metronomic Chemotherapy (MCT)

MCT is a suitable treatment option for selected MBC patients, first introduced to the clinic in international guidelines in 2017 for the treatment of advanced breast cancer. It consists in the continuous administration of low-dose chemotherapeutic agents with no or short regular treatment-free intervals [264]. MCT exerts its effects via immunomodulation, antiangiogenesis, and direct cytotoxic effects. Oral administration of MCT is safe, easy to handle, and allows for flexible drug dosing [265]. However, randomized controlled trials comparing MCT with conventional chemotherapic therapy are needed. The combination of MCT with other therapeutic interventions in BC is also currently under study [266].

## 5. Androgen Receptor Targeted Therapies

The AR, which belongs to the hormonal nuclear receptor superfamily, has been emerging as an interesting feature widely expressed in human BCs, both as a prognostic marker and for AR-targeting approaches [267,268]. However, its role in ER^+^ BC is controversial and has not yet been clearly established, thus constraining implementation of AR-directed therapies. Some authors report that AR suppression exerts potent antitumor activity in cancer [269], whereas other authors suggest that the activation of this receptor is responsible for the antitumor activity in ER^+^ BCs, including endocrine resistance [270]. AR has been considered a useful biomarker in BC, depending on the context of breast cancer subtypes [271]. It has also been shown to be predictive for the potential response to adjuvant hormonal therapy in ER+ BCs and neoadjuvant chemotherapy in TNBC and clinical trials using AR antagonists in BC are ongoing. Targeting AR, alone or combined with other therapeutic agents, provides alternatives to the existing therapies for BC, for instance enzalutamide and bicalutamide are two AR antagonists, which are in clinical trials to determine their efficacy alone to treat ER^+^ BC (NCT01889238 and NCT00468715, respectively). Moreover, a phase II trial (NCT02091960) that combined enzalutamide with the anti-HER2 antibody trastuzumab, in advanced AR+/HER2+/ER^−^ BC groups, indicated that about a quarter of the participants exhibit partial response or stable disease during the 24-week treatment period [272]. An ongoing phase II trial (NCT02750358) shows promising results in indicating AR^+^ patients’ tolerance to enzalutamide [267]. The combination of bicalutamide with the CDK4/6 selective inhibitors palbociclib and ribociclib is also under study in advanced AR^+^ TNBC (NCT02605486 and NCT03090165, respectively) [273]. On the other hand, AR agonists combined with standard-of-care agents enhanced therapeutic responses [270].

## 6. Immune Checkpoint Inhibitors (ICIs)

ICIs—which target the programmed cell death receptor 1 (PD-1) and programmed death ligand 1 (PD-L1)—have been widely explored in the field of BCs, including both early and advanced disease. The development of immunotherapy in BC is relatively slow as it is traditionally considered to be poorly immunogenic [274]. However, chemotherapy and ICIs act in a synergistic manner when administered together: chemotherapy triggers the release of cancer antigens by killing tumor cells, thus enhancing cancer antigens presentation by antigen-presenting cells [275]. In the ongoing NCT03036488 study, carried out on early TNBC patients, the percentage of a pathological complete response was considerably higher among patients who received pembrolizumab plus neoadjuvant chemotherapy than among those who received placebo plus neoadjuvant chemotherapy [276]. Recently, atezolizumab targeting PD-L1 in combination with paclitaxel was approved by the FDA for the treatment of TNBC [277].

## 7. Treatment of Breast Cancer Stem Cells

Cancer stem cells, or cancer initiating cells or tumor-initiating cells, are smaller cells within tumor that can repopulate cancer cells over a long period of time and maintain their ability to regenerate. Thus, they act as normal stem cells but in a diseased manner. The activation of 5′-adenosine monophosphate-activated protein kinase (AMPK) phosphorylates and targets several cellular pathways involved in cell growth and proliferation and is emerging as a target of choice in the treatment of cancer cells and cancer stem cells. For instance, paclitaxel involves the activation of AMPK during its therapeutic pathway [278,279]. Metformin, a drug commonly used to treat type 2 diabetes, has been proposed as a repositioned drug due to its ability to activate AMPK. Moreover, vanadium compounds are well-known PTP inhibitors and AMPK activators [280].

## 8. Natural Products

Plant-derived natural compounds are developing as a prospective therapeutic tool in cancer biology research due to their easy accessibility and cost-effectiveness [281]. The vinca alkaloids—vinblastine and vincristine—isolated from the Madagascar periwinkle plant *Catharanthus roseus* G. Don. (Apocynaceae), were the first natural products with potent antitumor properties for use in clinical cancer therapy. They interfere with the mitotic spindle apparatus, thus triggering cellular arrest in metaphase during mitosis leading to tumor cell death by apoptosis. These compounds are usually used in BC patients in combination chemotherapy or as a therapeutic option if other chemotherapy agents fail [282]. Vinorelbine and vinflunine are analogs of vinca alkaloids and act as antimitotics that bind tubulin, inhibit microtubule function and arrest mitosis [283]. Various natural agents—including curcumin, indol-3-carbinol, resveratrol, kaempferol, epigallocatechin gallate (EGCG), and genistein—are used in preclinical or clinical settings for the management of cancer [284]. Moreover, epidemiological data suggest that high nutritional intake of fruits and vegetables lower the risk of cancer [285].

## 9. Nanomedicines and Nanopharmaceuticals

Numerous papers evaluating a great variety of nanoformulations have been published, ranging from inorganic to organic nanoparticles, with high versatility, controllable size, and shape. The use of active targeted nanomedicine is an explored strategy to obtain the selective delivery of antineoplastics to cancer cells, thus reducing their adverse effects and increasing their efficacy. However, the toxicological effects of nanomaterials must be taken into account, even though some toxic effects are not yet completely known. It is mandatory to implement guidelines to standardize preclinical nanomedicine research [286]. In recent decades, several active targeted nanoformulations have been approved or reached clinical investigation with very good results. Besides ADC, which are probably the most promising nanomedicines, other active targeted nanosystems have reached the clinic for the treatment of BC including Abraxane^®^ and Nab-rapamicyn (albumin nanoparticles entrapping rapamycin) and various liposomes (MM-302, C225-ILS-Dox, and MM-310) loaded with doxorubicin or docetaxel and coated with ligands targeting HER-2 and other receptors, such as Ephrin 2. Abraxane consists of albumin nanoparticles of around 130 nm entrapping paclitaxel and received approval by the FDA and EMA in the 2000s for the treatment of different cancers, including BC. Nowadays, it is used in MBC after failure of combination chemotherapy and when standard treatment including an anthracycline is inadequate. Recently, it has been approved in combination with atezolizumab in PD-L1 overexpressing unresectable locally advanced tumors or metastatic TNBC. Several clinical studies are ongoing to evaluate the efficacy of combination therapy with abraxane [287].

## 10. Conclusions

BC is the most commonly diagnosed cancer in women worldwide with more than 2 million new cases in 2020. The way to view BC is widely and constantly changing, mostly after the further characterization of its molecular hallmarks that includes, for instance, the immunohistochemical markers ER, PR, HER2; the genomic markers *BRCA1*, *BRCA2*, and *PIK3CA*; and immunomarkers such as PD-L1. The neoadjuvant combination therapy represents a standard strategy for treating HER2^+^ and triple negative BCs, radiotherapy is still a valid cornerstone for BCs therapy, endocrine therapy and chemotherapy have been the treatment of choice, mostly in the last 10 years, for ER+ tumors, whereas targeted therapies using CDK4/6, PI3K, PARP inhibitors, and anti-PD-L1 immunotherapy, have been adopted for MBC. It is clear that the availability of wide treatment options reflects the complexity of BC today. It must also be considered that an optimal therapy should take into account the tumor subtype, cancer stage, and patients’ compliance and preferences. Thus, the future research in BC needs to be focused not only on new drugs discovery and repurposing, but mostly on the personalization of the therapy and the “precision medicine”. As an example, different approved agents (namely, PARP or PI3K inhibitors, amongst many others) work selectively against tumors exhibiting certain biomarkers or mutations. Some targeted therapy drugs, as monoclonal antibodies, control cancer cells and boost the immune system as well; thus, they can be considered as immunotherapy, reaching different parts of the body and are useful against MBC. Moreover, the use of antibody–drug conjugates for the treatment of TNBC as sacituzumab govitecan, targeting TROP2 successfully improved the progression-free survival and overall. New endocrine agents, as selective ER downregulators, have been developed to prevent or overcome the endocrine resistance, which is due, for instance, to ESR1 mutations. Finally, it is important to highlight the concept of finding the appropriate combination of treatments for each type of BC, because cancer cells could become resistant to targeted therapies over time. In this review, we presented the classification of BCs and listed the most commonly adopted therapies, shedding light on the complexity and the constant evolution of classic and emerging strategies and ongoing clinical trials.

## Figures and Tables

**Figure 1 ijms-24-03643-f001:**
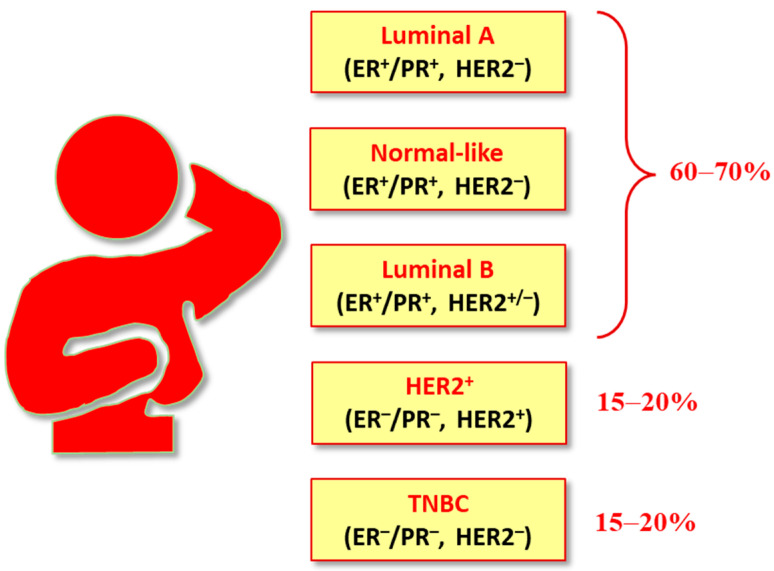
Molecular subtypes of BCs.

**Table 1 ijms-24-03643-t001:** Characteristics of adjuvant and neoadjuvant therapies.

Characteristics	Adjuvant	Neoadjuvant
**Definition**	Adjuvant therapy is the treatment after tumor removal	Neoadjuvant therapy is the treatment before tumor removal
**Advantages**	Can help prevent cancer recurrence and kills the remaining cancerous cells	Helps shrink the size of tumors making them easier to cut out
**Disadvantages**	The treatments can further weaken the organism due to unpleasant side effects	Delay in surgical removal of the tumor could mean the cancer spread
**Cancer caused by genetic mutations**	Most often recommended because the risk of cancer recurrence is high where mutations are inherited	Not often recommended if cancer is not due to inherited mutations but is likely due to environmental factors
**Early insight into treatment plan**	Does not provide early insight into the effects of chemotherapy and radiation until after surgery	Useful in indicating how a patient does with certain chemotherapy or radiation treatments before surgery, allowing for changes to be made after surgery

**Table 2 ijms-24-03643-t002:** Endocrine drugs in therapy and/or in clinical study for BC.

Structure	Name	Class	Phase Study
** 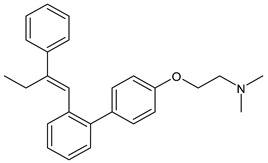 **	Tamoxifen (Nolvadex^®^, Kessar^®^, Nomafen^®^, Tamoxene^®^)	SERM	FDA approved
** 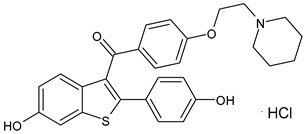 **	Raloxifene Hydrochloride (Evista^®^)	SERM	FDA approved
** 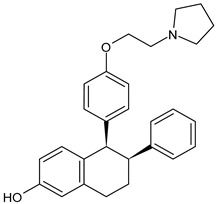 **	Lasofoxifene (Fablyn^®^)	SERM	FDA approved
** 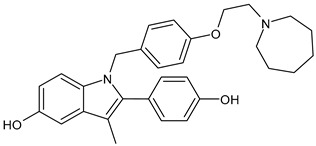 **	Bazedoxifene (Duavive^®^)	SERM	FDA approved
** 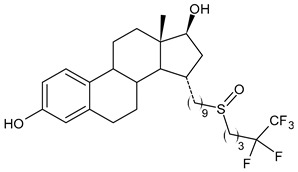 **	Fulvestrant (Faslodex^®^)	SERD	FDA approved
** 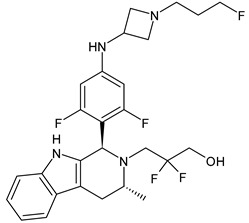 **	Giredestrant (GDC-9545)	SERD	Phase III
** 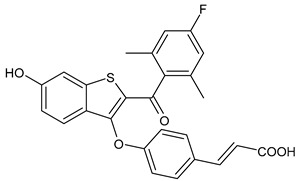 **	Rintodestrant (GIT48)	SERD	Phase I
** 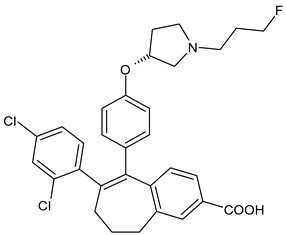 **	Amcenestrant (SAR439859)	SERD	Phase III
** 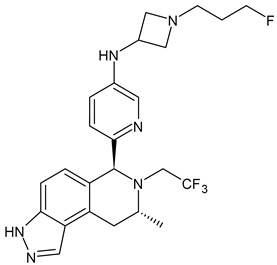 **	Camizestrant (AZD9833)	SERD	Phase III
** 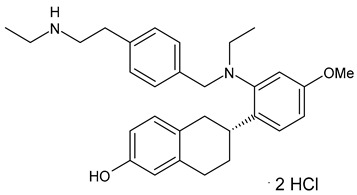 **	Elacestrant (RAD1901)	SERD	Phase III
** 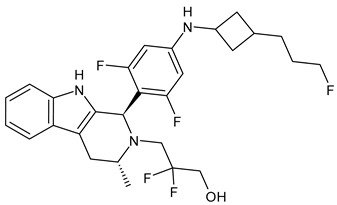 **	Imlunestrant (LY3484356)	SERD	Phase I
** 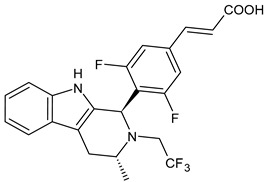 **	AZD9496	SERD	Phase I
** 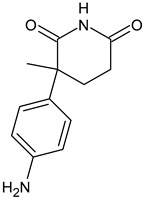 **	Aminoglutethimide (Orimeten)	AI First generation type II (non-steroidal)	FDA approved
** 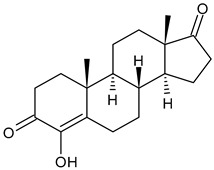 **	Formestane (Lentaron)	AI Second generation type I (steroidal)	FDA approved
** 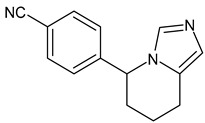 **	Fadrazole (Afema^®^)	AI Second generation type II (non-steroidal)	FDA approved
** 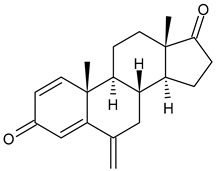 **	Exemestane (Aromasin^®^)	AI Third generation type I (steroidal)	FDA approved
** 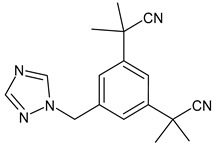 **	Anastrozole (Arimidex^®^)	AI Third generation type II (non-steroidal)	FDA approved
** 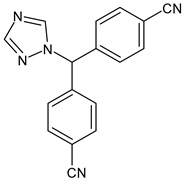 **	Letrozole (Femara^®^)	AI Third generation type II (non-steroidal)	FDA approved
** 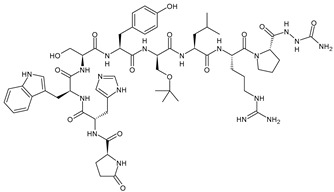 **	Goserelin	GnRH agonist	FDA approved
** 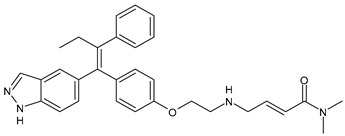 **	H3B-5942	SERCA	Precursor of SERCA H3B-6545
** 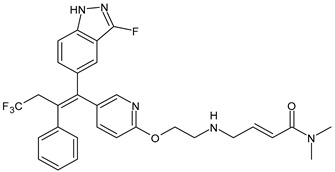 **	H3B-6545	SERCA	Phase II
** 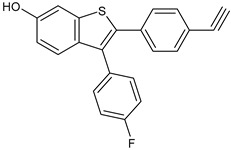 **	BMI-135	ShERPA	Phase I
** 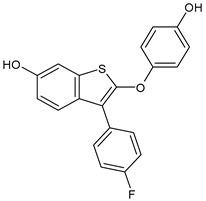 **	TTC-352	ShERPA	Phase I

**Table 3 ijms-24-03643-t003:** Chemotherapics in therapy and/or in clinical study for BC.

Structure	Name	Activity
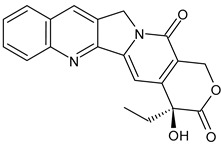	Camptothecin	TOPOI inhibitor
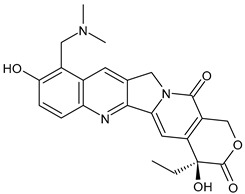	Topotecan	TOPOI inhibitor
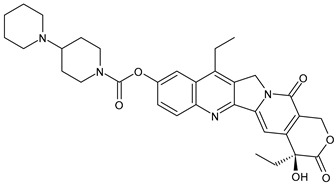	Irinotecan	TOPOI inhibitor
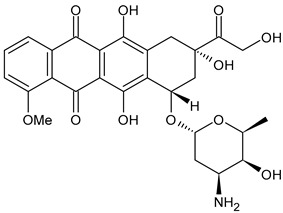	Adriamycin or doxorubicin	TOPOII inhibitor
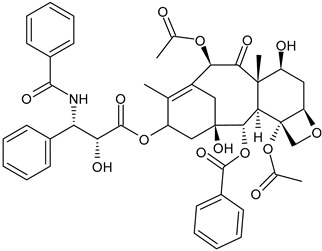	Paclitaxel	Tubulin stabilizer
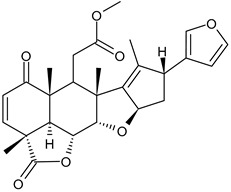	Nimbolide	Actin regulator
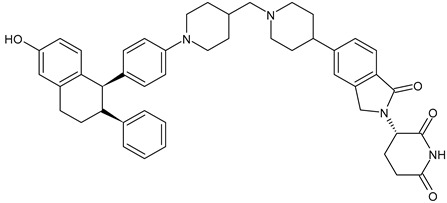	ARV-471	PROTACs

**Table 4 ijms-24-03643-t004:** Targeted therapies for BC.

Structure	Name	Class	Reference Number of Clinical Trials
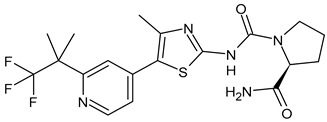	Alpelisib	PI3K inhibitor (selective PI3Kα)	NCT02437318
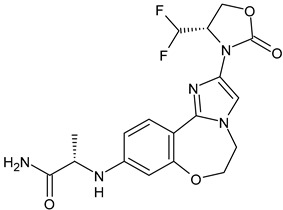	Inavolisib	PI3K inhibitor (selective PI3Kα)	NCT03006172
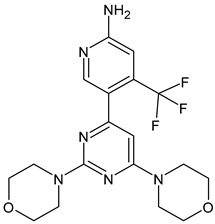	Buparlisib	PI3K inhibitor (pan-PI3K)	NCT01790932, NCT01629615
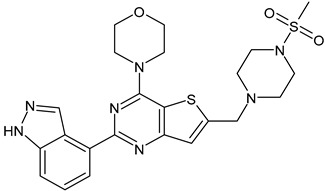	Pictilisib (GDC-0941)	PI3K inhibitor (pan-PI3K)	NCT01740336
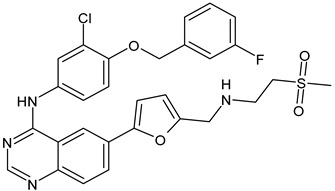	Lapatinib (GW-572016, Tyverb)	TKI	NCT03894410
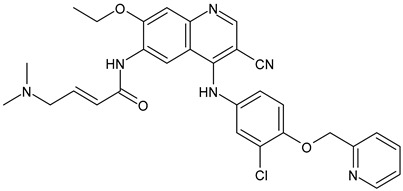	Neratinib (HKI-272) (Nerlynx™)	TKI	NCT01670877
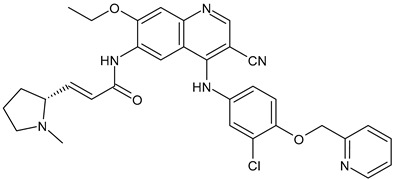	Pyrotinib	TKI	NCT03863223
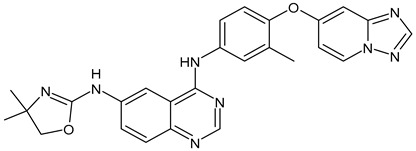	Tucatinib (Tukysa^®^)	TKI	NCT02614794
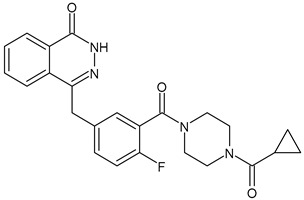	Olaparib (AZD-2281, Lynparza)	PARPi	NCT01034033
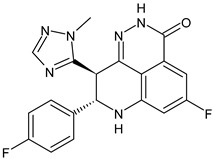	Talazoparib (BMN-673, Talzenna)	PARPi	NCT04039230
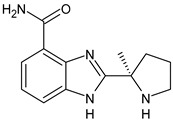	Veliparib (ABT-888)	PARPi	NCT01149083
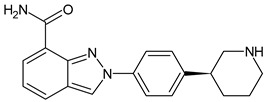	Niraparib (MK-4827)	PARPi	NCT04837209
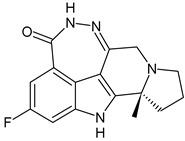	Pamiparib (BGB-290)	PARPi	NCT03575065
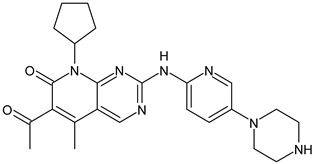	Palbociclib (Ibrance)	CDK4/6 inhibitor	NCT04711252
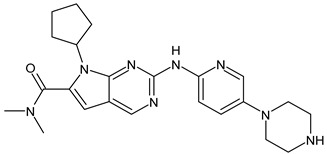	Ribociclib (Kisqali)	CDK4/6 inhibitor	NCT03577197
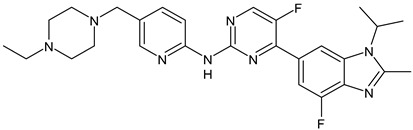	Abemaciclib (Verzenios)	CDK4/6 inhibitor	NCT02246621
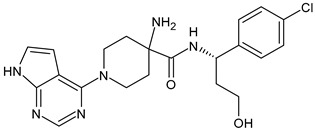	Capivasertib	Akt inhibitor	NCT03742102
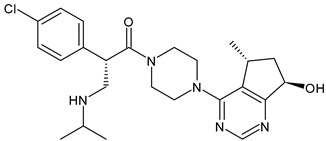	Ipatasertib	Akt inhibitor	NCT03337724, NCT04177108

## Data Availability

Not applicable.

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
