# Peer review of "Targeting Breast Cancer: An Overlook on Current Strategies"

_ijms, 2023, doi:10.3390/ijms24043643_

Round 1

Reviewer 1 Report

In this manuscript, the authors described various methods used for breast cancer treatment. They provided a comprehensive overview of this particular topic. As a general review, the content is sufficient. Overall, the manuscript is smooth and easy to understand. The authors can further improve their manuscript by supplementing it with more clinical information, such as the survival outcomes and cancer recurrence rate of the patients compared between treated and untreated patients during adjuvant setting. So, the readers will appreciate how the treatment improves patient management.

Below are some specific points the authors may consider:

1.       In table 2, it will be good to indicate which phases of clinical trials the drugs currently participated in or the drugs are already approved by the FDA for clinical usage.

2.       In table 4, it will be good to include the reference number/ID of the clinical trials. Therefore, readers will be easier to track the information and clinical aspects of the drugs.

3.       In section 4.2.5, the authors may consider mentioning the molecular subtypes of breast cancer employed in the studies to examine the effectiveness of the drugs. Alternatively, any biomarker indicates the suitability of the treatment.

4.       Several studies (e.g. Nature Medicine volume 27, pages 310–320 (2021); npj Breast Cancer volume 6, Article number: 47 (2020), etc) indicate the importance of androgen receptor in breast cancer. Targeting androgen receptor has been shown to benefit patients through monotherapy or in combination with other drugs. Several clinical trials have been conducted, for example, NCT03090165, NCT02605486, NCT02750358, etc. The authors may consider including information on androgen receptor in their review.

Author Response

REVIEWER 1

The authors can further improve their manuscript by supplementing it with more clinical information, such as the survival outcomes and cancer recurrence rate of the patients compared between treated and untreated patients during adjuvant setting. So, the readers will appreciate how the treatment improves patient management.

We agree with the referee. However, diverse therapies exist in adjuvant setting. They should be analyzed case by case in order to give correct survival outcomes and cancer recurrence rate details. In our opinion it could be an interesting cue for another review.

  1. In table 2, it will be good to indicate which phases of clinical trials the drugs currently participated in or the drugs are already approved by the FDA for clinical usage.

We are grateful to the reviewer for this suggestion. It was done.

  1. In table 4, it will be good to include the reference number/ID of the clinical trials. Therefore, readers will be easier to track the information and clinical aspects of the drugs.

We are grateful to the reviewer for this suggestion. It was done. The number of clinical trials regarding breast cancer were added.

  1. In section 4.2.5, the authors may consider mentioning the molecular subtypes of breast cancer employed in the studies to examine the effectiveness of the drugs. Alternatively, any biomarker indicates the suitability of the treatment.

We thank the reviewer for this suggestion. It was done.

  1. Several studies (e.g. Nature Medicine volume 27, pages 310–320 (2021); npj Breast Cancer volume 6, Article number: 47 (2020), etc) indicate the importance of androgen receptor in breast cancer. Targeting androgen receptor has been shown to benefit patients through monotherapy or in combination with other drugs. Several clinical trials have been conducted, for example, NCT03090165, NCT02605486, NCT02750358, etc. The authors may consider including information on androgen receptor in their review.

We thank the reviewer for this suggestion. Paragraph 5 (Androgen Receptor Targeted Therapies) was added.

Finally, other mistakes were corrected. We also found a duplicate in references (225 and 227). Ref 227 was deleted, and the other references renumbered.

Reviewer 2 Report

The review  "Targeting Breast Cancer: an overlook on current strategies" is an interesting scientific study on the treatment of such an important oncological issue as breast cancer. This review is a very detailed study of therapeutic methods along with the chemical formulas of the drugs used.      The Authors' choice of references, the mostly published within the last 10 years, is very important. This proves that the given data is up-to-date.            I rate this review positively due to the practical aspect. It can be used not only by the health service, but also by chemists involved in the synthesis of organic compounds used in cancer therapy.                                                 My review is supported by positive reviews of 281 publications in the References section.                                                                              Therefore, I propose to accept the review "Targeting Breast Cancer: an overlook on current strategies" in IJMS in its current form.

Author Response

  1. 02. 2023

REVIEWER 2

The review "Targeting Breast Cancer: an overlook on current strategies" is an interesting scientific study on the treatment of such an important oncological issue as breast cancer. This review is a very detailed study of therapeutic methods along with the chemical formulas of the drugs used. The Authors' choice of references, the mostly published within the last 10 years, is very important. This proves that the given data is up-to-date. I rate this review positively due to the practical aspect. It can be used not only by the health service, but also by chemists involved in the synthesis of organic compounds used in cancer therapy. My review is supported by positive reviews of 281 publications in the References section. Therefore, I propose to accept the review "Targeting Breast Cancer: an overlook on current strategies" in IJMS in its current form.

We are grateful to reviewer 2 for the comment.